# The Human 8-oxoG DNA Glycosylase 1 (*OGG1*) *Ser*326*Cys* Polymorphism in Infertile Men

**DOI:** 10.3390/biomedicines12102286

**Published:** 2024-10-09

**Authors:** César Antonio González-Díaz, María Antonieta Suárez-Souto, Elvia Pérez-Soto, Modesto Gómez-López, Jacobo Esteban Munguía-Cervantes, Nadia Mabel Pérez-Vielma, Virginia Sánchez-Monroy

**Affiliations:** 1Escuela Superior de Medicina, Instituto Politécnico Nacional, Mexico City 11340, Mexico; cgonzalezd@ipn.mx (C.A.G.-D.); mgomezlo@ipn.mx (M.G.-L.); 2Escuela Nacional de Medicina y Homeopatía, Instituto Politécnico Nacional, Mexico City 07320, Mexico; elvperezs@ipn.mx; 3Centro de Nanociencias y Micro y Nanotecnologías, Instituto Politécnico Nacional, Mexico City 07738, Mexico; jmunguia@ipn.mx; 4Centro Interdisciplinario de Ciencias de la Salud Unidad Santo Tomás, Instituto Politécnico Nacional, Mexico City 11340, Mexico; nperezv@ipn.mx

**Keywords:** infertility, 8-oxoguanine DNA glycosylase 1, semen

## Abstract

Background/Objectives: 8-hydroxy-2′-deoxyguanosine (8-OHdG) is a form of oxidative DNA damage caused by oxidative stress (OS), which is considered a major factor in male infertility. The cellular defense system against 8-OHdG involves base excision repair (BER) with the enzyme 8-Oxoguanine DNA glycosylase 1 (OGG1). However, studies on the single-nucleotide polymorphism (SNP) OGG1 *Ser*326*Cys* have demonstrated that the *Cys*326*Cys* genotype could be the cause of an increment in oxidative DNA damage. In this study, the OGG1 *Ser*326*Cys* polymorphism and its effect on DNA oxidation were evaluated in 118 infertile men. Methods: Polymorphic screening was performed using TaqMan allelic discrimination assays, and oxidative DNA damage was evaluated through the quantification of 8-OHdG and total antioxidant capacity (TAC); in addition, electrical bioimpedance spectroscopy (EBiS) measurements were used as a reference for different electrical properties associated with 8-OHdG concentrations. Results: The detected *Cys* (G) allele frequency (0.4) was higher compared to the allele frequency reported in the “Allele Frequency Aggregator” (ALFA) and “Haplotype Map” (HapMap) projects for American populations (0.21–0.29), suggesting that the *Cys* (G) allele carrier could be a factor associated with American infertile populations. The values of 8-OHdG were twofold higher in carriers of the *Cys*326*Cys* (GG) genotype than the other genotypes and, in concordance, the TAC levels were threefold lower in *Cys*326*Cys* (GG) genotype carriers compared to the other genotypes. Moreover, the EBiS magnitude exhibited potential for the detection of different oxidative damage in DNA samples between genotypes. Conclusions: The *Cys*326*Cys* (GG) genotype is associated with oxidative DNA damage that could contribute to male infertility.

## 1. Introduction

Infertility is a worldwide reproductive health problem affecting approximately 15% of couples [1]. Male infertility accounts for 60% of infertility problems [2], with semen abnormality being a common cause [1]. Several studies have been conducted to elucidate the etiology of male infertility and have shown that factors such as infection, inflammation signals, failure in antioxidant enzymatic activity, and lifestyle are potential contributors to an increase in reactive oxygen species (ROS) in semen, which leads to oxidative stress (OS) [1]. OS is an imbalance between ROS and antioxidant activity and is considered a leading factor in male infertility [3,4,5,6]. ROS are highly toxic molecules that attack cellular lipids, proteins, and DNA, causing alterations in their structure and function. DNA damage in semen is a major concern as it can become fixed in the genome as a permanent mutation [7]. Genomic damage caused by oxidative attack in semen has been evidenced by apoptotic alterations, chromatin decondensation, and DNA fragmentation [8,9,10,11].

DNA base lesions, which are chemical modifications to the base of a nucleotide, are the most common type of genomic damage [7]. The oxidation of DNA bases by ROS can occur directly in the genomic DNA strands or indirectly in the nucleotide pool, with subsequent incorporation into DNA during replication or repair, generating mutations [12]. Of the four DNA bases, guanine is the most susceptible to oxidation due to its low oxidation potential [13]. Its oxidation products are the main forms of oxidative damage. 8-hydroxy-2′-deoxyguanosine (8-OHdG) is one of the major forms of oxidative DNA damage and is a useful marker of cellular oxidative stress in human cells, including spermatozoa [14,15]. Significant elevation in the levels of 8OHdG in human spermatozoa is highly correlated with DNA fragmentation [14] and alterations in sperm parameters [15]. Moreover, 8-OHdG is considered to be a highly mutagenic DNA lesion if it is not repaired by forming a stable base pair with adenine, resulting in the transversion mutations G:C-T:A during DNA synthesis [16].

The cellular defense system against 8-OHdG involves base excision repair (BER) with the enzyme 8-Oxoguanine DNA glycosylase 1 (OGG1). The OGG1 enzyme, encoded by the OGG1 gene (*OGG1*), is located on chromosome 3p26.2 and plays a key role in the BER pathway in recognizing oxo-G:C base pairs, catalyzing both the expulsion of oxoG and the cleavage of the DNA backbone [17]. BER deficiency through polymorphism in the *OGG1* gene has been documented; the most common polymorphism in *OGG1* is a C:G transversion at nucleotide 1245, resulting in the substitution of cysteine (*Cys*) for serine (*Ser*) at codon 326 (*Ser*326*Cys*) [18]. The *Ser*326*Cys* polymorphism has been linked to increased oxidative DNA damage in different types of human cancer including breast, lung, head and neck [19], cervical, ovarian, and endometrial [20]. However, reports analyzing semen from infertile men are limited and mainly involve Asian and Spanish populations [21,22,23]. Reports that included infertile subjects with fertile subjects as controls in Chinese [21], Taiwanese [22], and Spanish [23] populations found a higher incidence of men carrying the *Cys*326 *OGG1* allele in the infertile groups compared to the fertile groups. Moreover, it was observed that carriers of the *Cys*326 *OGG1* allele were more vulnerable to oxidative damage [21,22,23], generating morphologically abnormal spermatozoa [23]. In this study, the *OGG1 Ser*326C*ys* polymorphism in the semen of a group of infertile men and its effect on DNA oxidation were evaluated.

## 2. Materials and Methods

### 2.1. Study Population

This cross-sectional study included 118 infertile male volunteers, defined according to the World Health Organization’s (WHO) specification of the failure to achieve a pregnancy after 12 months or more of regular unprotected sexual intercourse [24]. The participants were recruited during infertility consultations between January 2016 and November 2019 at the Hospital Militar de Especialidades de la Mujer y Neonatología of the Secretaría de la Defensa Nacional (SEDENA) in Mexico City. The inclusion criteria included men attending the hospital for conjugal infertility investigations. Young, healthy, physically fit military men undergoing drug therapy and those with undescended testes, varicocele, or other structural abnormalities were excluded. Ethical approval and informed consent were granted by the hospital’s Institutional Human Research Ethical Committee (19-CI-09-016-025). Individuals were informed of the research objectives, agreed to participate voluntarily, and signed an informed consent form. The Helsinki Declaration and the Official Mexican Standard (NOM-012-SSA3-2012) [25] were applied.

### 2.2. Sample Collection

Semen samples were obtained from all participants via masturbation after two to seven days of sexual abstinence. The collected samples were allowed to liquefy for 30 min at 37 °C. Semiology tests were performed, and the sperm parameters were evaluated according to the current World Health Organization (WHO) guidelines [26]. The semen samples were centrifuged at 300× *g* for 10 min, seminal plasma was obtained and stored at −20 °C until analysis, and cell pellets were separated and stored at 4 °C for nucleic acid extraction in the following 48 h.

### 2.3. Polymorphism Genotyping of OGG1 Gene

DNA extraction from the samples was performed using the DNeasy Blood and Tissue Kit (Cat. 69504 Qiagen NV, Hilden, Germany) according to the manufacturer’s protocol. The polymorphic screening was performed using TaqMan allelic discrimination assays (Cat. C___3095552_1_Applied Biosystems, Foster City, CA, USA) using a Techne Prime Thermal Cycler (Antylia Scientific, Vernon Hills, IL, USA). To confirm the detection of the polymorphism, polymerase chain reaction (PCR) using Platinum Hot Start PCR 2X Master Mix (Cat. 13000013 Invitrogen, Life Technologies, Carlsbad, CA, USA) was performed with the following specific primers: forward 5′TTCCACCTCCCAACACTGTCA-3′ and reverse 5′TGCCTGGCCTTTGAGGTAGT3′. The primers were designed with Primer Express 3.0.1 software (Applied Biosystems, Waltham, MA, USA). After PCR, the products were purified using ExoSAP-IT^TM^ (Cat. 78201.1ML, Applied Biosystems Foster City, CA, USA) and sequenced using the ABI Big Dye Terminator v3.1 Cycle Sequencing Kit (Cat. 4337455, Applied Biosystems, Foster City, CA, USA) in the ABI Prism 3130 Genetic Analyzer automated sequencer (Applied Biosystems, Foster City, CA, USA).

### 2.4. Oxidative DNA Damage Evaluation

8-hydroxy-2′deoxyguanosine (8-OHdG) quantification: The levels of 8-OHdG in DNA were measured using the competitive assay with enzyme-linked immunoassay (ELISA) method. The DNA/RNA Oxidative Damage (High Sensitivity) ELISA Kit (Cat. 589320, Cayman Chemical, Ann Arbor, MI, USA) was used. The absorbance of the samples was determined using a microplate reader (Thermo Fisher Scientific, Whalthman, MA, USA) at a wavelength of 420 nm. The levels of 8-OHdG were calculated using a standard curve, with a coefficient of determination (R^2^) value of 0.98. The assay has a working range of 10.3–3000 pg/mL and a sensitivity of approximately 30 pg/mL.

Total antioxidant capacity (TAC) quantification: The non-enzymatic TAC of seminal plasma was evaluated using a modified method described by Benzie and Strain [27], called the ferric reducing ability of plasma (FRAP) assay [28]. The FRAP working solution was prepared with 10 volumes of 300 mmol/L acetate buffer, pH 3.6; 1 volume of 10 mmol/L 2,4,6-tripyridyl-S-triazine in 40 mmol/L HCl; and 1 volume of 20 mmol/L FeCl_3_·6H_2_O, and was warmed to 37 °C. Next, 30 μL of seminal plasma, 90 μL of distilled water, and 30 μL of each standard solution (FeSO_4_·7H_2_O: 1000, 750, 500, 250, 100 μM) were transferred to an Eppendorf tube (2 mL) and 900 μL of the FRAP working solution was added. The mixture was heated to 37 ◦C for 10 min and the absorbance values of blank, standard solutions and samples were determined using a spectrophotometer (Thermo Fisher Scientific, Waltham, MA, USA) at a wavelength of 593 nm. The results were corrected for dilution and are expressed as μmol FeSO_4_/L. The standard curve was linear between 100 and 1000 μmol/L FeSO_4_ (coefficient of determination, R^2^ = 0.99). The results are expressed in μmol/L and compared with FeSO_4_ as standard. The measurements were performed in triplicate.

Electrical bioimpedance spectroscopy (EBiS) detection: The EBiS detection was conducted using a system that measured 2 μL volume samples through a 10 × 10 mm microelectrode interdigitated array, as well as a bioimpedance analyzer (Sciospec ISX-3, Sciospec Instruments GmbH, Leipziger, Bennewitz, Germany) in the frequency range of 100 Hz to 10 MHz in 126 logarithmically spaced steps.

### 2.5. Statistical Analysis

All statistical analyses were performed using the software SPSS version 23 (SPSS Inc., Chicago, IL, USA). Genotypes and alleles were expressed as frequency. Observed and expected genotypes were evaluated using a χ^2^ test. Descriptive statistics was used for the comparison of seminal parameters and 8-OHdG. The data are expressed as means ± standard deviation (SD) for normal distribution and median [25th, 75th] percentiles for abnormal distribution. To compare groups, a one-factor analysis of variance (ANOVA) parametric statistical test was used with a post hoc Tukey test, and the Kruskal–Wallis one-way ANOVA nonparametric test (K–W test) was used post hoc with varying effect sizes (eta square η^2^p; small ≥ 0.01, medium ≥ 0.06, large ≥ 0.14). A *p*-value ≤ 0.05 was considered statistically significant. 

## 3. Results

In this study, *OGG1 Ser*326*Cys* polymorphism was evaluated in a population of 118 men selected during infertility consultations. The clinical characteristics of the men are summarized in Table 1. They were males aged between 27 and 47 years, and the majority presented failure to achieve a pregnancy after 2 years or more. A high percentage (42%) reported having two or more sexual partners. With respect to semen parameters, a high percentage of the patients (81/118, 47%) exhibited teratozoospermia, with the resulting median of normal morphology altered for all participants.

The detected genotypes of *OGG1* were *Ser*326*Ser*, *Ser*326*Cys*, and *Cys*326*Cys* (Figure 1).

The genetic and allelic frequencies are summarized in Table 2. No differences are shown in the genotype distribution of the observed and expected genotypes, according to Hardy–Weinberg law [29] (χ^2^ = 0.2416, *p* = 0.623).

The evaluation of oxidative DNA damage via 8-OHdG quantification shows significant differences between genotypes. The values of 8-OHdG are approximately twofold higher in patients with the *Cys*326*Cys* (GG) genotype compared to the other genotypes (Figure 2).

The TAC quantification values in the seminal plasma were threefold lower in patients with the *Cys*326*Cys* (GG) genotype than for men with other genotypes (*Ser*326*Ser* (CC), 854 μmol/L (341.8–1015.3); *Ser*326*Cys* (CG), 638 μmol/L (295.8–987.8) and 238.29 μmol/L (54.7–413.8), *p <* 0.005) (Figure 3).

The EBiS of the representative DNA samples with *Ser*326*Ser* (CC), *Ser*326*Cys* (CG), and *Cys*326*Cys* (GG) genotypes was plotted in magnitude and their spectra displayed. EBiS measurements are expressed as the mean value of three assays with the corresponding standard error. Differences in electrical properties were detected between genotypes (Figure 4). The patient with the *Cys*326*Cys* (GG) genotype showed lower impedance compared to the other genotypes.

The semen quality parameters exhibit no differences between genotypes (Table 3).

## 4. Discussion

In this study, the effects of *OGG1 Ser*326*Cys* polymorphism in semen oxidative DNA damage were evaluated in a population of males with infertility. Because the etiology of DNA damage is complex [1,4], all of the studied patients were young, healthy, physically fit military men with a similar occupation and diet, which suggested that these factors did not influence the results. However, is important to note that other factors not considered in this study can lead to oxidative DNA damage and male infertility, such as immunological alterations due to stress and other clinical disorders reported in military communities [1,4,30]. Moreover, the patients were residents of the Mexico City Metropolitan Area, which is the largest urban area in North America and is also part of Latin America, characterized by severe exposure to air pollutants [31]. The residence variable is important, because some recently published studies have demonstrated an association between living in regions with high levels of pollution, such as organic pesticides, organic solvents, heavy metals, and volatile organic compounds (VOCs), and alterations in sperm morphology, sperm motility, sperm count, and protamine/histone ratios in young healthy males [32]. Such exposure to pollution can reduce the ability of sperm’s nuclear basic proteins to protect DNA from oxidative damage [33]. Moreover, 42% of the studied population reported having more than one sexual partner; this characteristic contributes to the acquisition of pathogens that affect semen quality. For example, asymptomatic infertile men infected with Human Papillomavirus and *Chlamydia trachomatis* exhibited high lipid peroxidation and 8-OHdG levels and low values of antioxidant capacity in sperm plasm, evidenced by oxidative stress, DNA damage, and altered seminal parameters [28]. In addition, the studied population exhibited a high frequency of teratozoospermia (69%), an abnormality of the sperm that has been associated with oxidative stress [3,8].

The polymorphism analysis showed the stability of the population, which was confirmed according to the Hardy–Weinberg principle, indicating an absence of selection mutation and migration [29]. Previous studies have shown that the *Cys*326 *OGG1* allele has a higher incidence in infertile men compared to fertile men in Asian and Spanish populations [21,22,23]. However, in this work, non-fertile men were used as a control group. This is because it is difficult to obtain fertile volunteers since they do not visit the clinic unless they have symptoms, thus representing an important limitation of this study. This work found a higher *Cys* (G) allele frequency (0.4) compared with the values reported in the allele frequency database by the “Allele Frequency Aggregator” (ALFA) project for general Latin American populations (0.21–0.29) [34] and the Haplotype map (HapMap) project for general American populations (0.22) [35], suggesting that the *Cys* (G) allele carrier could play a major role in cases of infertility in the American population. These projects determine the common patterns of DNA sequence variation in the human genome by characterizing variants and their allele frequencies. The projects include data from millions of subjects of geographically diverse populations and compare the frequency across populations. These high-volume and diverse data are freely available in the public domain and offer enormous potential for the identification of genetic factors that influence health and disease [34,35].

With respect to oxidation, we evaluated the levels of 8-OHdG as the major form of oxidative damage [14]. Similar to the findings in this work, Chen et al. [22] reported an approximately twofold increase in the level of 8OHdG in patients with the *Cys*326*Cys* (GG) genotype with a large effect size, highlighting the clinical importance of this marker and suggesting that polymorphism is associated with low enzyme activity. In addition, we evaluated TAC levels because excessive oxidative stress may contribute to deficiencies in the available antioxidant protection, thus affecting DNA integrity [36]. Similar to the findings in another report [22], the data showed approximately threefold lower TAC levels in carriers of the *Cys*326*Cys* (GG) genotype compared to the patients with other genotypes, with a large effect size, which demonstrates a higher OS state and less effective antioxidant defense in patients with the *Cys*326*Cys* (GG) genotype.

EBiS is a technique used for examining the electrical properties of biological material, and has shown great potential for DNA characterization [37]. Given the nature of DNA’s electronegative charge, it has the ability to conduct an electric current once influenced by a potential, and this ability depends on the molecular structure [38]. Thus, we evaluated the potential use of EBiS as an indicator of the degree of DNA oxidation damage in representative samples of DNA with *Ser*326*Ser* (CC), *Ser*326*Cys* (CG), and *Cys*326*Cys* (GG) genotypes and different 8-OHdG concentrations. The findings indicate that the EBiS magnitude parameter exhibited sensitivity in terms of the detection of differences between genotypes. The sample with the GG genotype exhibited lower impedance, likely caused by a higher level of 8-OHdG that could be an indicator for the change in the conformation of DNA structure, ionic diffusion, conductivity, and other electrical properties [39]. In contrast, the sample with the CC genotype exhibited higher impedance, in agreement with a lower 8-OHdG level. However, the differences detected using EBiS could also be explained by increments in DNA strand breaks, which has been demonstrated in other DNA samples analyzed before, during, and after radiation exposure, leading to differences in the EBiS spectra [40], similar to the findings reported in this study. In addition, DNA strand breaks caused by an increase in oxidative damage to the DNA have previously been correlated with high 8-OHdG levels in infertile men [14,41].

In relation to the quality parameters of semen, although other studies have reported a correlation between carriers of *Cys*326*Cys* (GG) and the semen parameters of concentration [22] and morphology [21], no differences were observed in this study between semen quality parameters and genotypes. The low sample size could explain the different results obtained in this study compared to previous studies; however, the high frequency of teratozoospermia (69%) in the study population also could be a reason.

Taken together, these results support previous evidence that indicates an association between the *Cys*326*Cys* (GG) genotype and the low repair activity of OGG1, which contributes to increased oxidation, affecting DNA fragmentation, fertilization, embryonic development, the incidence of miscarriages, and morbidity in the offspring [42,43]. However, several studies have reported improved semen parameters such as sperm motility, morphology, and concentration [44], and significant improvements in sperm redox status such as good correlation with pregnancy [5,45]. The data presented here could contribute to improving therapies for these patients, such as antioxidant therapy supplementation, which could reduce the time taken to conceive.

Finally, the data presented here should be interpreted with caution; the limitations of this study include the lack of a control group, the size of the sample, and the failure to consider other factors that can also lead to oxidative DNA damage, such as smoking and alcohol consumption and stress levels.

## 5. Conclusions

The results of this study indicate that carrying the *Cys*326*Cys* (GG) genotype contributes to increased DNA oxidation, which is suspected to be an important factor in male infertility. The findings suggest that antioxidant therapy may be useful to protect sperm from oxidative damage. Further research is essential to determine the effect of antioxidants in this population.

## Figures and Tables

**Figure 1 biomedicines-12-02286-f001:**
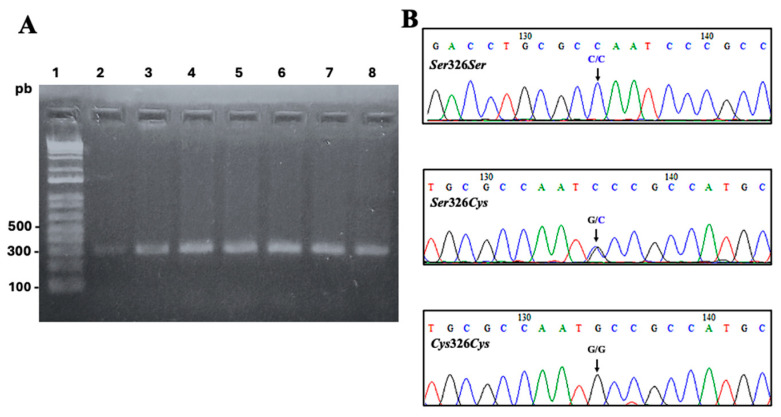
*OGG1 Ser*326*Cys* polymorphism detection. (**A**) Agarose (2%) gel electrophoresis of amplicons from representative samples. Lane 1: DNA ladder 100 pb. Lanes 2–8: 300 pb amplicons of *OGG1* obtained from representative samples. (**B**) Representative electropherograms showing the sequence of *OGG1* amplicons with genotypes detected.

**Figure 2 biomedicines-12-02286-f002:**
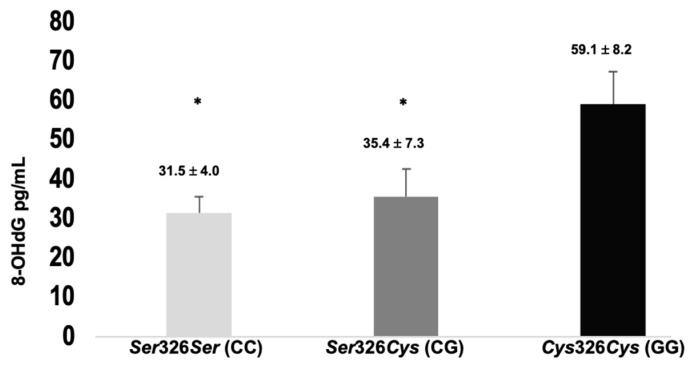
8-Hydroxi-2′deoxyguanosine (8-OHdG) levels by genotype. Data are expressed as mean ± standard deviation and were analyzed using one-way ANOVA. * Genotype GG vs. other genotype groups shows significant difference (significant difference is defined as *p* ≤ 0.05) and a large effect size (η^2^p = 0.184).

**Figure 3 biomedicines-12-02286-f003:**
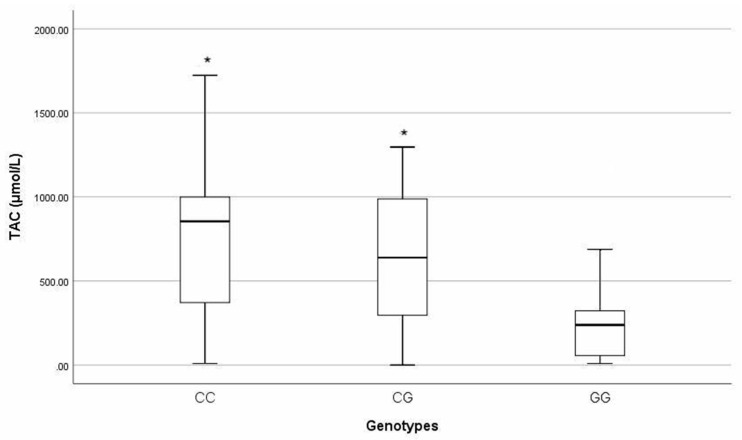
Total antioxidant capacity (TAC) levels by genotype. Data are expressed as median and compared using Kruskal–Wallis test. * Genotype GG vs. other genotype groups shows significant difference. Significant difference is defined as *p* ≤ 0.05 and a large effect size (η^2^p = 0.102).

**Figure 4 biomedicines-12-02286-f004:**
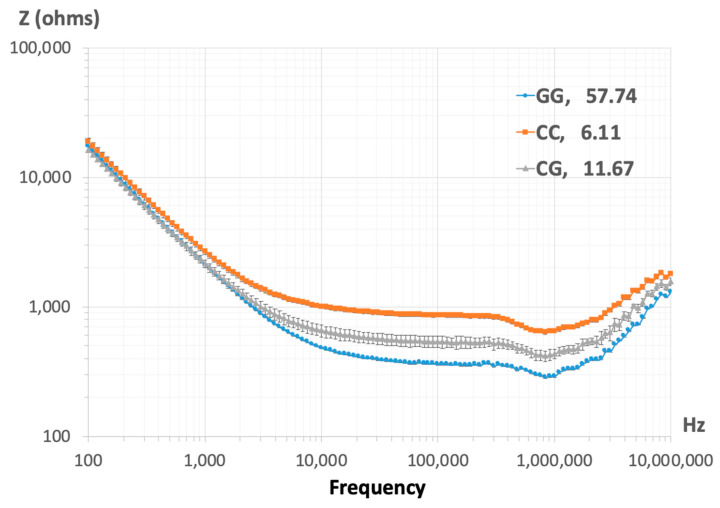
EBiS magnitude spectra for representative samples with the *Cys*326*Cys* (GG), *Ser*326*Ser* (CC), and *Ser*326*Cys* (CG) genotypes. The levels of 8-OHdG are indicated by genotype in pg/mL.

**Table 1 biomedicines-12-02286-t001:** Characteristics of the population studied.

Age (y)	30 (26, 34)
Failure to achieve a pregnancy	
after 1 year *n* (%)	25 (21)
after 2 years or more *n* (%)	93 (79)
Number of sexual partners	
1 *n* (%)	68 (58)
2 *n* (%)	31 (26)
3 or more *n* (%)	19 (16)
Occupation	Military
Semen Parameters	
Volume (mL)	3.2 (2.2, 3.8)
Total sperm count (10^6^/ejaculate)	75.0 (46, 146)
Total motility (%)	44 ± 22
Normal morphology (%)	3.0 (2.0, 5.0)

Normally distributed data are reported as mean and standard deviation; skewed data are reported as median [25th, 75th] percentiles.

**Table 2 biomedicines-12-02286-t002:** Genetic and allelic frequencies of *OGG1 Ser*326*Cys* polymorphisms in study population.

Genotype	*Ser*326*Ser* (CC) *N* (%)	*Ser*326*Cys* (CG) *N* (%)	*Cys*326*Cys* (GG) *N* (%)
Observed Frequency	44 (37)	54 (46)	20 (17)
Expected Frequency	43 (36)	56 (47)	19 (16)
Allelic Frequency			
Allele C	71 (60)		
Allele G	47 (40)		

Genotype distribution was in Hardy–Weinberg equilibrium (χ^2^ = 0.2416 *p* = 0.623).

**Table 3 biomedicines-12-02286-t003:** Semen analysis and its association with OGG1 polymorphism.

Semen Parameters	*Ser*326*Ser* (CC) *n* = 44	*Ser*326*Cys* (CG) *n* = 54	*Cys*326*Cys* (GG) *n* = 20	Normal Value WHO (6th Edition)	*p*
Volume (mL)	3.1 (2.3, 4.0)	2.8 (2.5, 3.7)	2.4 (1.9, 4.7)	1.4 (1.3–1.5)	0.47
Total sperm count (10^6^/ejaculate)	68.5 (34, 122)	95 (53, 161)	64 (42, 96)	39 (35–40)	0.15
Total motility (%)	43 ± 22	43 ± 22	47 ± 23	42 (40–43)	0.79
Normal morphology (%)	3.0 (2.0, 4.0)	3.0 (2.0, 5.0)	2.0 (2.0, 5.0)	4 (3.9–4)	0.56

Normally distributed data are reported as mean and standard deviation; skewed data are reported as median [25th, 75th] percentiles. Significant difference is defined as *p* ≤ 0.05.

## Data Availability

The data that support the findings of this study are available on reasonable request from the corresponding author.

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
