# Peer review of "The Human 8-oxoG DNA Glycosylase 1 (OGG1) Ser326Cys Polymorphism in Infertile Men"

_biomedicines, 2024, doi:10.3390/biomedicines12102286_

Round 1

Reviewer 1 Report

Comments and Suggestions for Authors

This work is well designed and well written. The methodology is adequate. However, it needs minor revisions as follow:

1.      The sample size estimation was not mentioned in material and methods.

2.      It is necessary that characteristics of study group added to the manuscript.

3.      It would be interesting to increase the discussion of possible explanations for the differences that were observed.

Author Response

Dear editor

 Thank you very much for your opportunity, we have analyzed and reviewed point by point the comments and corrections, undoubtedly our manuscript will improve notably.

Revisor 1. This work is well designed and well written. The methodology is adequate. However, it needs minor revisions as follow:page1image41296512

Comment 1. The sample size estimation was not mentioned in material and methods. 

Response 1. The appreciation of the reviewers is correct, thank you very much for your observation. We did not estimate a sample size, we recruited patients who attended the clinic and met the inclusion criteria over a period of time, indicate in material and methos line 82 as follow:

This cross-sectional study included 118 infertile male volunteers, defined according to the World Health Organization’s (WHO) specification of the failure to achieve a pregnancy after 12 months or more of regular unprotected sexual intercourse [24]. The participants were recruited during infertility consultations between January 2016 and November 2019 at the Hospital Militar de Especialidades de la Mujer y Neonatología of the Secretaría de la Defensa Nacional (SEDENA) in Mexico City. The inclusion criteria included men attending the hospital for conjugal infertility investigations. Young, healthy, physically fit military men undergoing drug therapy and those with undescended testes, varicocele, or other structural abnormalities were excluded. Ethical approval and informed consent were granted by the hospital's Institutional Human Research Ethical Committee (19-CI-09-016-025). Individuals were informed of the research objectives, agreed to participate voluntarily, and signed an informed consent form. The Helsinki Declaration and the Official Mexican Standard (NOM-012-SSA3-2012) were applied. 

Comment 2. It is necessary that characteristics of study group added to the manuscript.

Response 2. Thank you for the observations, we now include in results section line 160, Table 1 with characteristics of population studied, as follow:

In this study, OGG1 Ser326Cys polymorphism was evaluated in a population of 118 men selected during infertility consultations. The clinical characteristics of the men are summarized in Table 1. They were males aged between 27 and 47 years, and the majority presented failure to achieve a pregnancy after 2 years or more. A high percentage (42%) reported having two or more sexual partners. With respect to semen parameters, a high percentage of the patients (81/118, 47%) exhibited teratozoospermia, with the resulting median of normal morphology altered for all participants

Table 1 Characteristics of the population studied

Age (y)

30 (26,34)

Failure to achieve a pregnancy

after 1 year n (%)

25 (21)

after 2 years or more n (%)

93 (79)

Number of sexual partners

1 n (%)

68(58)

2 n (%)

31(26)

3 or more n (%)

19(16)

Occupation

Military

Semen Parameters

Volume (mL)

3.2 (2.2,3.8)

Total sperm count (106/ejaculate)

75.0 (46, 146)

Total motility (%)

44 ± 22

Normal morphology (%)

3.0 (2.0, 5.0)

Normally distributed data are reported as mean and standard deviation; skewed data are reported as median [25th, 75th] percentiles.

Comment 3. It would be interesting to increase the discussion of possible explanations for the differences that were observed.

Response 3. Thank you for the observations, we now increase the Discussion Section line 211 as follow:

In this study, the effects of OGG1 Ser326Cys polymorphism in semen oxidative DNA damage were evaluated in a population of males with infertility. Because the etiology of DNA damage is complex [1,4], all of the studied patients were young, healthy, physically fit military men with a similar occupation and diet, which suggested that these factors did not influence the results. However, is important to note that other factors not considered in this study can lead to oxidative DNA damage and male infertility, such as immunological alterations due to stress and other clinical disorders reported in military communities [1,4, 29]. Moreover, the patients were residents of the Mexico City Metropolitan Area, which is the largest urban area in North America and is also part of Latin America, characterized by severe exposure to air pollutants [30]. The residence variable is important, because some recently published studies have demonstrated an association between living in regions with high levels of pollution, such as organic pesticides, organic solvents, heavy metals, and volatile organic compounds (VOCs), and alterations in sperm morphology, sperm motility, sperm count, and protamine/histone ratios in young healthy males [31]. Such exposure to pollution can reduce the ability of sperm’s nuclear basic proteins to protect DNA from oxidative damage [32]. Moreover, 42% of the studied population reported having more than one sexual partner; this characteristic contributes to the acquisition of pathogens that affect semen quality. For example, asymptomatic infertile men infected with Human Papillomavirus and Chlamydia trachomatis exhibited high lipid peroxidation and 8-OHdG levels and low values of antioxidant capacity in sperm plasm, evidenced by oxidative stress, DNA damage, and altered seminal parameters [27]. In addition, the studied population exhibited a high frequency of teratozoospermia (69%), an abnormality of the sperm that has been associated with oxidative stress [3,8].

Thank you very much for your evaluation and opportunity

Reviewer 2 Report

Comments and Suggestions for Authors

The paper presented for review is devoted to the study of the distribution of the polymorphism OGG1 Ser326Cys in infertile men and its effect on DNA oxidation. The work was performed on 118 infertile men. There are the following questions and suggestions about work: 1. In the abstract, in the materials and methods section, the number of men in the studied sample should be indicated. 2. Why did the authors not study a control sample of men without infertility to confirm that Ser326Cys polymorphism is a risk factor for infertility in the study group (as suggested in the abstract)? 3. How did the authors determine the sample size required for the study? What is the power of this study? 4. The paper should provide information on the biomedical characteristics (age, BMI, etc.) and the main risk factors for infertility (if studied) in general in the studied sample and in comparison in groups of men with infertility with different genotypes (for example, in Tables 1 or 2). This will also make it possible to understand whether any covariates should be used in statistical analysis. 5. At the end of the discussion of the data obtained, it is recommended to indicate the limitations of this study. 6. What prospects can there be for the clinical use of the data obtained (there may be a correction of treatment in individuals with certain genotypes, etc.)? In general, the results obtained are interesting, convincing, and make a significant contribution to the development of ideas about the mechanisms of infertility in men.

Author Response

Dear editor

 Thank you very much for your opportunity, we have analyzed and reviewed point by point the comments and corrections, undoubtedly our manuscript will improve notably

Revisor 2.

The paper presented for review is devoted to the study of the distribution of the polymorphism OGG1 Ser326Cys in infertile men and its effect on DNA oxidation. The work was performed on118 infertile men. There are the following questions and suggestions about work: 

Comment 1. In the abstract, in the material and methods section, the number of men in the studied samples should be indicated.

Response 1. Thank you for the observations, we now include in the abstract section line 20 and materials and methods section line 82 the number of men in the study sample as follow:

Abstract section. In this study, the OGG1 Ser326Cys polymorphism and its effect on DNA oxidation were evaluated in 118 infertile men.

Materials and methods section. This cross-sectional study included 118 infertile male volunteers,

Comment 2. Why did the authors not study a control sample of men without infertility to confirm that Ser 326Cys polymorphism is a risk factor for infertility in the study group. (As suggest in the abstract)?

Response 2. The appreciation of the reviewers is correct, thank you very much for your observation. We comment in the discussion section line 238 as follow:

Previous studies have shown that the Cys326 OGG1 allele has a higher incidence in infertile men compared to fertile men in Asian and Spanish populations [21-23]. However, in this work, non-fertile men were used as a control group. This is because it is difficult to obtain fertile volunteers since they do not visit the clinic unless they have symptoms, thus representing an important limitation of this study. This work found a higher Cys (G) allele frequency (0.4) compared with the values reported in the allele frequency database by the “Allele Frequency Aggregator” (ALFA) project for general Latin American populations (0.21-0.29) [33] and the Haplotype map (HapMap) project for general American populations (0.22) [34], suggesting that the Cys (G) allele carrier could play a major role in cases of infertility in the American population. These projects determine the common patterns of DNA sequence variation in the human genome by characterizing variants and their allele frequencies. The projects include data from millions of subjects of geographically diverse populations and compare the frequency across populations. These high-volume and diverse data are freely available in the public domain and offer enormous potential for the identification of genetic factors that influence health and disease [33,34].

Comment 3. How did the authors determine the sample size required for the study? What is the power of this study?  

Response 3. The appreciation of the reviewers is correct, thank you very much for your observation. We did not estimate a sample size, we recruited patients who attended the clinic and met the inclusion criteria over a period of time. We now described in material and methods line 82 about sample size estimation as follow:

This cross-sectional study included 118 infertile male volunteers, defined according to the World Health Organization’s (WHO) specification of the failure to achieve a pregnancy after 12 months or more of regular unprotected sexual intercourse [24]. The participants were recruited during infertility consultations between January 2016 and November 2019 at the Hospital Militar de Especialidades de la Mujer y Neonatología of the Secretaría de la Defensa Nacional (SEDENA) in Mexico City. The inclusion criteria included men attending 

hospital for conjugal infertility investigations. Young, healthy, physically fit military men undergoing drug therapy and those with undescended testes, varicocele, or other structural abnormalities were excluded. Ethical approval

and informed consent were granted by the hospital's Institutional Human Research Ethical Committee (19-CI-09-016-025). Individuals were informed of the research objectives, agreed to participate voluntarily, and signed an informed consent form. The Helsinki Declaration and the Official Mexican Standard (NOM-012-SSA3-2012) were applied.

In relation to size effect of genotype we include the value in addition to p value in material and methods line156, results, and discussion sections as follow: 

Descriptive statistics was used for the comparison of seminal parameters and 8-OHdG. The data are expressed as means ± standard deviation (SD) for normal distribution and median [25th, 75th] percentiles for abnormal distribution. To compare groups, a one-factor analysis of variance (ANOVA) parametric statistical test was used with post hoc Tukey’s test, and the Kruskal–Wallis one-way ANOVA nonparametric test (K–W test) was used with post hoc and effect size (eta square η2p; small ≥ .01, medium ≥ .06, large ≥ .14). A p-value ≤ 0.05 was considered statistically significant.

3. Results lines 189 and 197

Figure 2. 8-Hydroxi-2’deoxyguanosine (8-OHdG) levels by genotype. Data are expressed as mean ± standard deviation and were analyzed using one-way ANOVA. *Genotypes GG vs. other genotype groups show significant difference (significant difference is defined as p0.05) and a large effect size (η2p=.184).

Figure 3. Total antioxidant capacity (TAC) levels by genotype. Data are expressed as median and compared using Kruskal–Wallis test. *Genotypes GG vs. other genotypes groups show significant difference. Significant difference is defined as p ≤ 0.05 and a large effect size (η2p=.102).

4. Discussion line 255

With respect to oxidation, we evaluated the levels of 8-OHdG as the major form of oxidative damage [14]. Similar to the findings in this work, Chen et al. [22] reported an approximately twofold increase in the level of 8OHdG in patients with the Cys326Cys (GG) genotype with a large effect size, highlighting the clinical importance of this marker and suggesting that polymorphism is associated with low enzyme activity. In addition, we evaluated TAC levels because excessive oxidative stress may contribute to deficiencies in the available antioxidant protection, thus affecting DNA integrity [35]. Similar to the findings in another report [22], the data showed approximately threefold lower TAC levels in carriers of the Cys326Cys(GG) genotype compared to the patients with other genotypes, with a large effect size, which demonstrates a higher OS state and less effective antioxidant defense in patients with the Cys326Cys (GG) genotype.

Comment 4. The paper should provide information on the biomedical characteristics (age, BMI, etc) and the main risk factors for infertility (if study) in general in the studied sample and in comparison, in groups of men with. Infertility with different genotypes (for example, in Tables 1 or 2). This. Will also make it possible. To understand whether. Any covariates should be used in statistical analysis

Response 4: Thank you for the observations, we now include Table 1 line 160 with characteristics of population studied, as follow:

In this study, OGG1 Ser326Cys polymorphism was evaluated in a population of 118 men selected during infertility consultations. The clinical characteristics of the men are summarized in Table 1. They were males aged between 27

and 47 years, and the majority presented failure to achieve a pregnancy after 2 years or more. A high percentage (42%) reported having two or more sexual partners. With respect to semen parameters, a high percentage of the patients (81/118, 47%) exhibited teratozoospermia, with the resulting median of normal morphology altered for all participants.

Table 1 Characteristics of the population studied

Age (y)

30 (26,34)

Failure to achieve a pregnancy

after 1 year n (%)

25 (21)

after 2 years or more n (%)

93 (79)

Number of sexual partners

1 n (%)

68(58)

2 n (%)

31(26)

3 or more n (%)

19(16)

Occupation

Military

Semen Parameters

Volume (mL)

3.2 (2.2,3.8)

Total sperm count (106/ejaculate)

75.0 (46, 146)

Total motility (%)

44 ± 22

Normal morphology (%)

3.0 (2.0, 5.0)

Normally distributed data are reported as mean and standard deviation; skewed data are reported as median [25th, 75th] percentiles.

In addition, we increase discussion section line 212 respect to characteristics as follow:

In this study, the effects of OGG1 Ser326Cys polymorphism in semen oxidative DNA damage were evaluated in a population of males with infertility. Because the etiology of DNA damage is complex [1,4], all of the studied patients were young, healthy, physically fit military men with a similar occupation and diet, which suggested that these factors did not influence the results. However, is important to note that other factors not considered in this study can lead to oxidative DNA damage and male infertility, such as immunological alterations due to stress and other clinical disorders reported in military communities [1,4, 29]. Moreover, the patients were residents of the Mexico City Metropolitan Area, which is the largest urban area in North America and is also part of Latin America, characterized by severe exposure to air pollutants [30]. The residence variable is important, because some recently published studies have demonstrated an association between living in regions with high levels of pollution, such as organic pesticides, organic solvents, heavy metals, and volatile organic compounds (VOCs), and alterations in

sperm morphology, sperm motility, sperm count, and protamine/histone ratios in young healthy males [31]. Such exposure to pollution can reduce the ability of sperm’s nuclear basic proteins to protect DNA from oxidative damage [32]. Moreover, 42% of the studied population reported having more than one sexual partner; this characteristic contributes to the acquisition of pathogens that affect semen quality. For example, asymptomatic infertile men infected with Human Papillomavirus and Chlamydia trachomatis exhibited high lipid peroxidation and 8-OHdG levels and low values of antioxidant capacity in sperm plasm, evidenced by oxidative stress, DNA damage, and altered seminal parameters [27]. In addition, the studied population exhibited a high frequency of teratozoospermia (69%), an abnormality of the sperm that has been associated with oxidative stress [3,8]

Comment 5. At the end of the discussion of the data obtained, it is recommended to indicate the limitations of this study.

Response 5: The appreciation of the reviewers is correct, thank you very much for your observation. We now indicate the limitations of this study in the discussion section line 296 as follow:

Finally, the data presented here should be interpreted with caution; the limitations of this study include the lack of a control group, the size of the sample, and the failure to consider other factors that can also lead to oxidative DNA damage, such as smoking and alcohol consumption and stress levels.

Comment 6. What prospects can there be for the clinical use of the data obtained (there may be a correction of treatment in individuals with certain genotypes, etc)?

Response 6: The appreciation of the reviewers is correct, thank you very much for your observation. We now indicate the clinical use of this study in the discussion section line 293 as follow:

Taken together, these results support previous evidence that indicates an association between the Cys326Cys (GG) genotype and the low repair activity of OGG1, which contributes to increased oxidation, affecting DNA fragmentation, fertilization, embryonic development, the incidence of miscarriages, and morbidity in the offspring [41,42]. However, several studies have reported improved semen parameters such as sperm motility, morphology, and concentration [43], and significant improvements in sperm redox status such as good correlation with pregnancy [5,44]. The data presented here could contribute to improving therapies for these patients, such as antioxidant therapy supplementation, which could reduce the time taken to conceive

Thank you very much for your evaluation and opportunity

Reviewer 3 Report

Comments and Suggestions for Authors

In the article titled “The Human 8-oxoG DNA Glycosylase 1 (OGG1) Ser326Cys Polymorphism in Infertile Men” the authors evaluated the OGG1 Ser326Cys polymorphism in infertile men group and its effect on DNA oxidation. I think that I can reconsider the possibility of publication after a major revision.

My questions are the following:

·        Better describe the selection criteria for recruited subjects. E.g. were they drinkers? smokers? did they use drugs? did they exercise? All confounding factors must be eliminated

·        On what basis have these subjects been defined as infertile? On the basis of the spermiogram?

·        The controls, i.e. the fertile men, how many are there? and how are they defined as fertile?

·        How long were the samples stored at 4°C before nucleic acid extraction? was whole semen stored? or spermatozoa?

·        Line 114: “The 8- hydroxy-2’deoxyguanosine (8-OHdG) quantification”. This sentence seems a title

·        Place subscripts where needed in the text

·        The data in this work are few and the work has many limitations. Try to arrange the data better and better explain the usefulness of each experimental approach also in relation to the other approaches used

·        The authors state that certain factors such as pollution have not been taken into account, but this factor is very important. At least cite some work, such as the following:

·         10.1186/s10020-023-00776-6

·        10.3390/ijerph191711023

These works show how certain pollutants can cause oxidative damage in the sperm DNA

·        The results must be commented on and an explanation must also be given for the effect of these substances on the alterations observed. A molecular mechanism must also be better hypothesized

·        try to correlate the results obtained with each other

·        Better define the limitations of this study

·        A revision of English is necessary

Comments on the Quality of English Language

 Extensive editing of English language required.

Author Response

Dear editor

Thank you very much for your opportunity, we have analyzed and reviewed point by point the comments and corrections, undoubtedly our manuscript will improve notabl

Revisor 3
In the article titled “The Human 8-oxoG DNA Glycosylase 1 (OGG1) Ser326Cys Polymorphism in Infertile Men” the authors evaluated the OGG1 Ser326Cys polymorphism in infertile men group and its effect on DNA oxidation. I think that I can reconsider the possibility of publication after a major revision.

Comment 1. Better describe the selection criteria for recruited subjects. E.g. were they drinkers? smokers? did they use drugs? did they exercise? All confounding factors must be eliminated

Response 1: The appreciation of this reviewers is correct, thank you very much for your observation we explain with more detail in material and methods section line 82 as follow:

This cross-sectional study included 118 infertile male volunteers, defined according to the World Health Organization’s (WHO) specification of the failure to achieve a pregnancy after 12 months or more of regula

unprotected sexual intercourse [24]. The participants were recruited during infertility consultations between January 2016 and November 2019 at the Hospital Militar de Especialidades de la Mujer y Neonatología of the  Secretaría de la Defensa Nacional (SEDENA) in Mexico City. The inclusion criteria included men attending the hospital for conjugal infertility investigations. Young, healthy, physically fit military men undergoing drug therapy and those with undescended testes, varicocele, or other structural abnormalities were excluded. Ethical approval and informed consent were granted by the hospital's Institutional Human Research Ethical Committee (19-CI-09-016-025). Individuals were informed of the research objectives, agreed to participate voluntarily, and signed an informed consent form. The Helsinki Declaration and the Official Mexican Standard (NOM-012-SSA3-2012) were applied.

Comment 2. On what basis have these subjects been defined as infertile? On the basis of the spermiogram?

Response 2. Thank you very much for your observation we explain in material and methods section line 82 as follow:

This cross-sectional study included 118 infertile male volunteers, defined according to the World Health Organization’s (WHO) specification of the failure to achieve a pregnancy after 12 months or more of regular unprotected sexual intercourse [24].

Comment 3.  The controls i.e. the fertile men, how many are there? and how are they defined as fertile?

Response 3: Thank you for the observations, this work non-fertile men as control group were evaluated, however now explain in the discussion section line 238 the databases used for comparing frequency in similar populations as follow:

However, in this work, non-fertile men were used as a control group. This is because it is difficult to obtain fertile volunteers since they do not visit the clinic unless they have symptoms, thus representing an important limitation of this study. This work found a higher Cys (G) allele frequency (0.4) compared with the values reported in the allele frequency database by the “Allele Frequency Aggregator” (ALFA) project for general Latin American populations (0.21-0.29) [33] and the Haplotype map (HapMap) project for general American populations (0.22) [34], suggesting that the Cys (G) allele carrier could play a major role in cases of infertility in the American population. These projects determine the common patterns of DNA sequence variation in the human genome by characterizing variants and their allele frequencies. The projects include data from millions of subjects of geographically diverse populations and compare the frequency across populations. These high-volume and diverse data are freely available in the public domain and offer enormous potential for the identification of genetic factors that influence health and disease [33,34].

Comment 4: How long were the samples stored at 4°C before nucleic acid extraction? was whole semen stored? or spermatozoa?

Response 4: Thank you very much for your observation We now described about stored samples in the material and methods line 96 as follow:

Semen samples were obtained from all participants via masturbation after two to seven days of sexual abstinence. The collected samples were allowed to liquefy for 30 min at 37 °C. Semiology tests were performed, and the sperm parameters were evaluated according to the current World Health Organization (WHO) guidelines [25]. The seme

samples were centrifuged at 300× g for 10 min, seminal plasma was obtained and stored at −20 â—¦C until analysis, and cell pellets were separated and stored at 4°C for nucleic acid extraction in the following 48h.

Comment 5 Line 114: “The 8- hydroxy-2’deoxyguanosine (8-OHdG) quantification”. This sentence seems a title. Place subscripts where needed in the text

Response 5. The appreciation of this reviewers is correct, thank you very much for your observation we now adjust the text using cursive letter

Comment 6.  The data in this work are few and the work has many limitations. Try to arrange the data better and better explain the usefulness of each experimental approach also in relation to the other approaches used

Response 6. Thank you very much for your observation, we organize data better:

  1. We described characteristic of population study and summarized in a new table
  2. We increase discussion section and support with 8 additional references

Comment 7. The authors state that certain factors such as pollution have not been taken into account, but this factor is very important. At least cite some work, such as the following:10.1186/s10020-023-00776-6 and 10.3390/ijerph191711023. These works show how certain pollutants can cause oxidative damage in the sperm DNA. The results must be commented on and an explanation must also be given for the effect of these substances on the alterations observed. A molecular mechanism must also be better hypothesizedº

Response 7. Thank you very much for your observation we now adjust the text in the discussion section line 215 as follow:

However, is important to note that other factors not considered in this study can lead to oxidative DNA damage and male infertility, such as immunological alterations due to stress and other clinical disorders reported in military communities [1,4, 29]. Moreover, the patients were residents of the Mexico City Metropolitan Area, which is the largest urban area in North America and is also part of Latin America, characterized by severe exposure to air pollutants [30]. The residence variable is important, because some recently published studies have demonstrated an association between living in regions with high levels of pollution, such as organic pesticides, organic solvents, heavy metals, and volatile organic compounds (VOCs), and alterations in sperm morphology, sperm motility, sperm count, and protamine/histone ratios in young healthy males [31]. Such exposure to pollution can reduce the ability of sperm’s nuclear basic proteins to protect DNA from oxidative damage [32]

Comment 8. Try to correlate the results obtained with each other

Response 8: Thank you very much for your observation we now adjust and increase all the discussion section line 212 as follow:

In this study, the effects of OGG1 Ser326Cys polymorphism in semen oxidative DNA damage were evaluated in a population of males with infertility. Because the etiology of DNA damage is complex [1,4], all of the studied patients were young, healthy, physically fit military men with a similar occupation and diet, which suggested tha these factors did not influence the results. However, is important to note that other factors not considered in this study can lead to oxidative DNA damage and male infertility, such as immunological alterations due to stress and other clinical disorders reported in military communities [1,4, 29]. Moreover, the patients were residents of the Mexico City Metropolitan Area, which is the largest urban area in North America and is also part of Latin America, characterized by severe exposure to air pollutants [30]. The residence variable is important, because some recently published studies have demonstrated an association between living in regions with high levels of pollution, such as organic pesticides, organic solvents, heavy metals, and volatile organic compounds (VOCs), and alterations in sperm morphology, sperm motility, sperm count, and protamine/histone ratios in young healthy males [31]. Such exposure to pollution can reduce the ability of sperm’s nuclear basic proteins to protect DNA from oxidative damage [32]. Moreover,42% of the studied population reported having more than one sexual partner; this characteristic contributes to the acquisition of pathogens that affect semen quality. For example, asymptomatic infertile men infected with Human Papillomavirus and Chlamydia trachomatis exhibited high lipid peroxidation and 8-OHdG levels and low values of antioxidant capacity in sperm plasm, evidenced by oxidative stress, DNA damage, and altered seminal parameters [27]. In addition, the studied population exhibited a high frequency of teratozoospermia (69%), an abnormality of the sperm that has been associated with oxidative stress [3,8].

The polymorphism analysis showed the stability of the population, which was confirmed according to the Hardy–Weinberg principle, indicating an absence of selection mutation and migration [28]. Previous studies have shown that the Cys326 OGG1 allele has a higher incidence in infertile men compared to fertile men in Asian and Spanish populations [21-23]. However, in this work, non-fertile men were used as a control group. This is because it is difficult to obtain fertile volunteers since they do not visit the clinic unless they have symptoms, thus representing an important limitation of this study. This work found a higher Cys (G) allele frequency (0.4) compared with the values reported in the allele frequency database by the “Allele Frequency Aggregator” (ALFA) project for general Latin American populations (0.21-0.29) [33] and the Haplotype map (HapMap) project for general American populations (0.22) [34], suggesting that the Cys (G) allele carrier could play a major role in cases of infertility in the American population. These projects determine the common patterns of DNA sequence variation in the human genome by characterizing variants and their allele frequencies. The projects include data from millions of subjects of geographically diverse populations and compare the frequency across populations. These high-volume and diverse data are freely available in the public domain and offer enormous potential for the identification of genetic factors that influence health and disease [33,34].

With respect to oxidation, we evaluated the levels of 8-OHdG as the major form of oxidative damage [14]. Similar to the findings in this work, Chen et al. [22] reported an approximately twofold increase in the level of 8OHdG in patients with the Cys326Cys (GG) genotype with a large effect size, highlighting the clinical importance of this marker and suggesting that polymorphism is associated with low enzyme activity. In addition, we evaluated TAC levels because excessive oxidative stress may contribute to deficiencies in the available antioxidant protection, thus affecting DNA integrity [35]. Similar to the findings in another report [22], the data showed approximately threefold lower TAC levels in carriers of the Cys326Cys (GG) genotype compared to the patients with other genotypes, with a large effect size, which demonstrates a higher OS state and less effective antioxidant defense in patients with the Cys326Cys(GG) genotype.

EBiS is a technique used for examining the electrical properties of biological material, and has shown great potential for DNA characterization [36]. Given the nature of DNA’s electronegative charge, it has the ability to conduct an electric current once influenced by a potential, and this ability depends on the molecular structure [37]. Thus, we evaluated the potential use of EBiS as an indicator of the degree of DNA oxidation damage in representative samples of DNA with Ser326Ser (CC), Ser326Cys (CG), and Cys326Cys (GG) genotypes and different 8-OHdG concentrations. The findings indicate that the EBiS magnitude parameter exhibited sensitivity in terms of the detection of differences between genotypes. The sample with the GG genotype exhibited a lower impedance, likely caused by a higher level of 8-OHdG that could be an indicator for the change in the conformation of DNA structure, ionic diffusion, conductivity, and other electrical properties [38]. In contrast, the sample with the CC genotype exhibited a higher impedance, in agreement with a lower 8-OHdG level. However, the differences detected using EBiS could also be explained by increments in DNA strand breaks, which has been demonstrated in other DNA samples analyzed before, during, and after radiation exposure, leading to differences in the EBiS spectra [39], similar to the findings reported in this study. In addition, DNA strand breaks caused by an increase in oxidative damage to the DNA have previously been correlated with high 8-OHdG levels in infertile men [14,40].

In relation to the quality parameters of semen, although other studies have reported a correlation between carriers of Cys326Cys (GG) and the semen parameters of concentration [22] and morphology [21], no differences were observed in this study between semen quality parameters and genotypes. The low sample size could explain the different results obtained in this study compared to previous studies; however, the high frequency of teratozoospermia (69%) in the study population also could be a reason.

Taken together, these results support previous evidence that indicates an association between the Cys326Cys (GG) genotype and the low repair activity of OGG1, which contributes to increased oxidation, affecting DNA fragmentation, fertilization, embryonic development, the incidence of miscarriages, and morbidity in the offspring [41,42]. However, several studies have reported improved semen parameters such as sperm motility, morphology, and concentration [43], and significant improvements in sperm redox status such as good correlation with pregnancy [5,44]. The data presented here could contribute to improving therapies for these patients, such as antioxidant therapy supplementation, which could reduce the time taken to conceive.

Finally, the data presented here should be interpreted with caution; the limitations of this study include the lack of a control group, the size of the sample, and the failure to consider other factors that can also lead to oxidative DNA damage, such as smoking and alcohol consumption and stress levels.

Comment 9. Better define the limitations of this study

Response 9: Thank you very much for your observation. We now indicate the limitations of this study in the discussion section line 296 as follow:

Finally, the data presented here should be interpreted with caution; the limitations of this study include the lack of a control group, the size of the sample, and the failure to consider other factors that can also lead to oxidative DNA damage, such as smoking and alcohol consumption and stress levels

Comment 10. A revision of English is necessary

Response 10: The appreciation of the reviewers is correct, thank you very much for your observation. We now review of English.

Round 2

Reviewer 2 Report

Comments and Suggestions for Authors

The authors provided all necessary comments on the comments/suggestions made and made all reasonable adjustments to the article. The article is recommended for publication.

Reviewer 3 Report

Comments and Suggestions for Authors

The authors addressed all my questions. I accept the manuscript in the present form 

Comments on the Quality of English Language

Minor editing of English language required